# The Impact of University-Related Variables on Students' Perceived Employability and Mental Well-Being: An Italian Longitudinal Study

**Giovanni Schettino \*** , **Leda Marino and Vincenza Capone**

Department of Humanities, University of Naples Federico II, Via Porta di Massa 1, 80133 Naples, Italy; leda.marino@unina.it (L.M.); vincenza.capone@unina.it (V.C.)

\* Correspondence: giovanni.schettino@unina.it

**Abstract:** The COVID-19 outbreak has had a disruptive impact on the academic context and labor market. Indeed, the pandemic shock in such fields has been related to several changes with implications for young people's careers and well-being. This two-wave longitudinal study, conducted in Italy, aimed to explore the predictiveness of some individual and organizational factors on students' perceived employability and well-being. A total of 301 Italian students, aged between 18 and 33 (M = 20.63, SD = 1.99), completed a self-report questionnaire measuring career ambition, university reputation, university commitment, technostress related to technology-enhanced learning, perceived employability, and mental well-being at both time points. A path analysis showed that career ambition, university reputation, and organizational commitment positively predicted employability, which, in addition to such variables, positively affected well-being. In contrast, technostress was identified as a risk factor both for students' perceptions of finding a job and for their well-being. These findings provide a theoretical contribution to a better understanding of the factors involved in undergraduates' perceived employability and well-being. Moreover, they suggest the need to improve academic-related variables to enhance individuals' resources in coping with the pandemic challenges.

**Keywords:** employability; mental well-being; career ambition; university reputation; university commitment

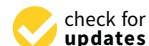



## 1. Introduction

The pandemic has significantly affected all aspects of our lives, including education and work, as highlighted by several reports about the employment rate. The latter has fallen by 0.7 percentage points in Europe: from 73.1% in 2019 to 72.4% in 2020 [1]. The situation is particularly worrying in Italy since it is ranked among the last European countries in terms of employment rates, with a percentage of 62.6 in 2020 [1].

Of particular concern is the finding regarding the youth unemployment rate: the average European rate is 15.4%. The highest rates were recorded in Spain (29.2%), Italy (28%), and Greece (39.1%) [2]. Along with unemployment, reduced well-being has become a serious concern during the pandemic. As underlined by studies on COVID-19 and mental health, anxiety symptoms have been reported to be moderate to severe by almost one-third of the population [3], and young people have been regarded as the highest-risk group for a range of mental illnesses [4,5].

These data show to what extent employability and well-being are fundamental concerns at the time of Coronavirus. A point of contact between such concepts might be identified in those variables that concern both the individuals and educational organizations. In support of this line of argument, it should be noted that students' feelings and attitudes toward school are linked to life satisfaction and well-being [6]. In addition, even in the era of such a health emergency, employability rates vary considerably with the level of education attained by young people [7].

Thus, consistent with previous studies [8], the crucial role of high-quality education in career success seems clear. Consequently, the mentioned evidence suggests analyzing in-depth the impact of learning contexts on young people's perception of gaining employment during the pandemic and their outcomes on individuals' well-being, also clarifying the underlying mechanism of this association.

### 1.1. Perceived Employability among University Students

Akkermans and colleagues [9] pointed out that COVID-19 could be regarded as a career shock in people's lives: "a disruptive and extraordinary event that is, at least to some degree, caused by factors outside the focal individual's control and that triggers a deliberate thought process concerning one's career" (p. 4). Due to this significant impact on the job market, it is necessary to study factors involved in individuals' perceptions of their career as well as the potential outcomes. This line of reasoning is aligned with the Event Systems Theory [10], which states that contextual (e.g., career shocks) and individual features (e.g., aspirations) jointly determine the outcomes of a career shock. This theory also says that intense events characterized by novelty, criticism, and disruption (such as the health emergency) are more likely to trigger career decisions. Furthermore, Ali and Mehreen [11] identified job-related self-efficacy and perceived employability as two valuable resources able to determine a better adaptation to career shocks. More specifically, the authors showed that such resources could mediate the relationship between career shock and protean career behaviors. Therefore, also among young people, the most vulnerable group of the population both before [12] and during the virus outbreak [13], perceived that employability could be identified as the critical factor in understanding their ability to cope with COVID-19 consequences on the labor market. Employability is generally defined as the ability to be employed [14]. The literature identified different conceptualizations of employability, considering both its subjective and objective aspects as well as their interplay relationship. Indeed, focusing on its objective dimensions, employability can be regarded as the individual's ability to obtain and maintain employment [15]. On the other hand, the subjective perspective of employability refers to perceived employability as the "individual's perception of his or her possibilities of obtaining and maintaining employment" [16] (p. 593) or to the appraisal of one's possibilities of getting new employment [17]. The increasing changes in the labor market have made it necessary to consider the development of employability throughout the individuals' entire life, and consequently to the definition of sustainable employability: the individual's long-term ability to work and remain employed [18]. In order to achieve this goal, a set of capabilities related to individuals and work is essential. In fact, when employees can convert their personal or contextual resources into capabilities, they are more likely to function well at work. As a result, this contributes to building sustainable employability and experiencing well-being [19]. Consequently, sustainable employability is not a personal characteristic but the result of a complex interaction between individuals, their job, and the social/organizational context [20,21].

Sustainable employability is established starting from when the individual is involved in university studies. Thus, in the case of undergraduates, perceived employability can be conceived as "the result of an internal evaluation process by the individual student of his/her personal capital (knowledge, abilities, skills, and traits acquired through formal education or experiential learning) as well as his/her external conditions (reputation of university, degree subject, and state of the labor market)" [22] (p. 1). These factors increase the likelihood of being employed once the individual graduates [23]. To summarize, individuals' perceptions of the usefulness of their studies and of how well they prepare them influence their employability, which in turn is positively associated with career success. In support, as argued by Martin and colleagues [24], graduates' satisfaction with academic resources is a strong predictor of perceived employability, highlighting the crucial impact of the educational program on the construction of their career path.

*1.2. Antecedents of Self-Perceived Employability*

According to the Conservation of Resources Theory [25], people are pushed to protect and build resources, preventing potential losses. Therefore, by focusing on the labor market, employability can be regarded as a personal resource to improve well-being [26]. More specifically, literature [27–32] has widely recognized that individuals with a higher level of employability report increased vigor at work, lower job exhaustion, higher job satisfaction, and career success. As a result, employability can promote a sense of control over one's career, with positive impacts for employees facing a career transition (e.g., job loss and job search) [33]. Indeed, individuals higher in employability believe that they have a greater variety of career options and opportunities to achieve their goals. Such a belief can also translate into positive outcomes in terms of well-being [29].

Concerning university students, those who perceive themselves as being more employable report higher self-efficacy and are more likely to engage in job-seeking behaviors [34]. Thus, it is plausible to imagine that employability may be a helpful resource in preserving and fostering students' well-being during economically strained times such as the pandemic. The need to increase levels of employability leads us to examine its antecedents.

One antecedent may be recognized as career ambition, which was defined by Rothwell et al. [35] as a proxy for the perception of future career success. This factor is conceived as the individual's willingness and commitment to achieve their professional goals and the adoption of concrete actions to realize them and to take into account the importance of a career for personal self-realization. In addition to employability, career ambition is a strong predictor of successful career development [36] and protean career orientation [37]. Moreover, research has suggested that individuals higher in career ambition are more resilient to stress [38] and experience greater mental well-being [29]. They are confident in their work-related capabilities (supported by personal and external resources) to perform challenging goals. Locke and Latham's [39] Goal Setting theory also suggests that goals' characteristics, such as clarity, play a role in motivating and maintaining efforts to achieve them. In line with this literature, Ćurić Dražić et al. [40] indicated in career ambition a predictor of perceived employability. Following this reasoning, ambition could be conceived as an underlying mechanism that placed values on outcomes to achieve, linking the beliefs about students' personal efforts to career success. In addition to ambition, research has recognized two other crucial factors in developing perceived employability: reputation and commitment. The perception of university reputation is a fundamental asset in the labor market, the main factor shaping students' study choices. Graduates from reputable universities are more likely to obtain job opportunities than those from low-rated universities [41–43]. This link can be interpreted by assuming that the best universities attract students with better academic abilities. In addition, high-ranked universities may provide or improve employability skills required by organizations (i.e., job-related self-efficacy, commitment, and portable skills), resulting in better job performance [41]. Hence, the great importance placed on reputation may influence students' choices between different universities [44], as well as beliefs about their career to the point that reputation may affect students' mental well-being [45] and perceived employability, as demonstrated by Pitan and Muller [22]. Their results show that university reputation might determine undergraduates' perceived employability, both directly and indirectly, through experiential learning activities. The authors concluded that high-ranking universities attract and enjoy deep relationships with employers (e.g., career exhibitions, internship opportunities, or any graduate recruitment exercise). As a result, their students report a high level of perceived employability because they recognize the value of these activities for career success.

Organizational commitment is a further pillar of perceived employability. Employee commitment is conceived as a strong belief and acceptance of the organization's objectives and values, a willingness to make a significant effort on behalf of the organization, and a strong desire to remain within it [46]. Such employees' bond with their companies has consequences at the individual and organizational levels. Indeed, research has documented that employee commitment is negatively related to absenteeism [47], turnover [48], and

counterproductive behavior [49]. In addition, highly committed employees show better job satisfaction [48], motivation [50], job performance [51], and both mental and physical well-being [52,53]. As posited by the Conservation of Resources Theory [25,54], organizational commitment provides employees with the resources to ensure their responsibilities as members of the organization and satisfy their needs (e.g., affiliation, esteem, emotional support) [55]. Through such a process, organizational commitment positively impacts employees' well-being.

In the context of higher education, students' commitment to their university has been positively associated with ambition, academic self-efficacy, university reputation, and perceived employability [29,35,56,57]. Nevertheless, as far as we know, only a correlational study [29] has investigated the link between university commitment and students' mental well-being in the pandemic, reporting a positive correlation between them, so further studies are needed to ascertain the direction of such relationships.

### 1.3. Students' Technostress and Remote Learning

As potential risk factors for employability and mental health during an emergency, it is necessary to evaluate the implications of the wide adoption of remote learning during the crisis. It has been particularly challenging for students, resulting in several concerns such as technostress: a person's incapacity to deal with new technologies and change cognitive and social needs associated with technology usage [58,59]. Thus, people affected by technostress feel unable to deal with quick changes in information and communication technologies (ICT) due to the perception of a mismatch between demands and resources in ICT usage, leading to increased unpleasant psychophysiological activation and negative attitudes towards ICT [60]. It could be argued that such a rapid change has been the sudden decision taken by most universities [61], including Italian ones, to convert courses entirely from face-to-face to technology-enhanced learning (TEL), often without adequate resources [62–65]. This new way of learning usually requires more time, knowledge, and skills, and thus psychological pressures [3,66–68], than the traditional one. As a result, in the pandemic, undergraduates experienced technostress [3,69], which, in turn, has led them to suffer from exhaustion [70], anxiety, and depressive symptoms [71], as well as report poorer academic performance [72]. As claimed by the Person–Environmental (P–E) fit theory [73], every stress does not come separately from the individual or the environment but rather develops from maladaptation between them. Therefore, there is a need to consider demands and resources at both an individual and organizational level. Following this approach, several studies have investigated the potential negative impact of technostress in workplaces, considering both individual and organizational characteristics. Their findings have revealed that such "modern disease" [58] could manifest its effects in the form of higher job dissatisfaction [74], job burnout [75], and intention to quit [76], as well as decreased organizational commitment [77], productivity [78], job engagement [79], and well-being [80].

The mentioned consequences make it plausible to suppose a significant impact of technostress on individuals' professional careers. In this regard, Van Vuure et al. [81], using a sample of IT employees, demonstrated that perceived technologies overload and the relative experience of complexity, but not their intrusion into personal life, might represent important stressors due to their negative association with workability, vigor, and perceived employability. However, no study has investigated its effects on students' perceived employability to the best of our knowledge. Therefore, it would be interesting to test whether the technostress related to TEL could be a risk factor for university students' well-being and their perception of gaining a job during a time invaded by communication technologies [82].

### 1.4. Aim and Hypotheses

In light of the literature mentioned earlier, this two-wave study aims to identify the predictors of mental well-being and perceived employability during the third wave of

the COVID-19 pandemic (which occurred from March to May 2021 [83], registering over 46,500 deaths [84]) in Italian university students at the beginning (T1) and the end (T2) of the second part of an academic year (Figure 1). Consequently, we tested the role of the academic context at the time of Coronavirus in order to implement strategies useful to improve students' perceived employability and well-being, especially in relation to technostress that could be considered a psychosocial risk. To this aim, we measured students' levels of career ambition, perceived university reputation and commitment, technostress, employability, and mental well-being. We further hypothesized that as follows:

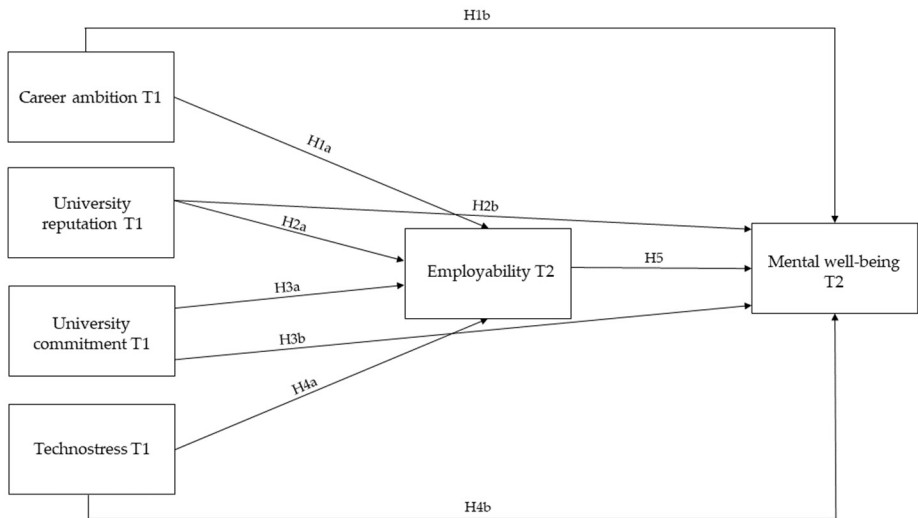

**Figure 1.** The hypothesized model.

**Hypothesis 1 (H1).** *T1 career ambition will positively affect T2 perceived employability (Hypothesis H1a) and T2 well-being (Hypothesis H1b).*

**Hypothesis 2 (H2).** *T1 university reputation will positively affect T2 perceived employability (Hypothesis H2a) and T2 well-being (Hypothesis H2b).*

**Hypothesis 3 (H3).** *T1 university commitment will positively affect T2 perceived employability (Hypothesis H3a) and T2 well-being (Hypothesis H3b).*

**Hypothesis 4 (H4).** *T1 technostress will negatively affect T2 perceived employability (Hypothesis H4a) and T2 well-being (Hypothesis H4b).*

**Hypothesis 5 (H5).** *T2 employability will positively affect T2 well-being.*

## 2. Materials and Methods

### 2.1. Participants

In March 2021, during the third wave of the pandemic, 800 university students attending a course in psychology at an Italian university were invited to take part in a longitudinal study to evaluate their perceptions of labor work at the time of Coronavirus. Among invited participants, five hundred and fifty Italian students (84% women), aged between 18 and 33 years (M = 20.63, SD = 1.99), signed the informed consent form and filled out the first questionnaire (T1; the response rate = 62.5%). After about three months, starting from 5 June 2021 (when the university course was ended), a total of 301 participants (82.7% women; $M_{age}$ = 20.91, SD = 1.93) completed the second questionnaire (T2), showing a drop-out rate of 54.7%. The questionnaire required a mandatory answer to each item. A personal code was generated by each participant in T1 and T2 for case-matching purposes. Most participants were unemployed (87.4%) and enrolled in the first academic year (68.8%). Statistical analyses were conducted only on participants who completed both question-

naires. Participation was anonymous, no incentive was given, and informed consent was obtained from all participants. All procedures followed were in accordance with the ethical standards and the Helsinki Declaration of 1975, as revised in 2000. Respondents were given the opportunity to withdraw from the study at any time.

### 2.2. Variables and Measures

The survey instrument was a self-report questionnaire that included, in the first section, questions regarding sociodemographics (age, gender, employment status) and participants' perceptions about their career, university, and well-being. The psychological variables under investigation were measured in the subsequent sections of the questionnaire.

Career ambition was assessed through Rothwell et al.'s [85] Ambition Scale, consisting of 6 items rated on a 5-point Likert scale from 1 (completely disagree) to 5 (completely agree). An example of such items is "I have clear goals for what I want to achieve in life". An index was calculated based on the average of items. Higher mean scores represent an indicator of greater career ambition. The reliability of the scale was $\alpha = 0.62$ at T1 and $\alpha = 0.72$ at T2.

Perceived university reputation was measured with a single item ("My university has an outstanding reputation in my field(s) of study") rated on a 5-point Likert scale ranging from 1 (strongly disagree) to 5 (strongly agree). Higher scores indicate a greater perception of one's university reputation.

Students' commitment to their university was measured with the University Commitment Scale [85]. This scale consists of 8 items (e.g., "I am proud to tell others I am at this university") rated on a 5-point Likert scale ranging from 1 (strongly disagree) to 5 (strongly agree). The items' mean was computed. Higher average scores reveal a stronger commitment between an individual and his/her university. Cronbach's alpha was 0.84 at T1 and 0.86 at T2.

Technostress related to technology-enhanced learning was assessed with the Technostress Scale for University Students in TEL [86]. Participants rated their agreement with eight statements (e.g., "I feel stressed to cope with the high demands of technology-enhanced learning with my current capability") on a 5-point Likert scale ranging from 0 (strongly disagree) to 4 (strongly agree). The items' average was calculated. Higher mean scores indicate a greater perceived technostress related to TEL. Cronbach's alpha was 0.92 both at T1 and 0.94 at T2.

Self-perceived employability was measured by adopting the Students' Self-Perceived Employability Scale [85] (SPES). The tool assesses individuals' perceptions of their value in the labor market and consists of 16 items (e.g., "I feel I could get any job so long as my as my skills and experience are reasonably relevant") evaluated on a 5-point Likert scale ranging from 1 (strongly disagree) to 5 (strongly agree). An index was built based on the mean of the items. Higher average scores indicate a greater belief that the individual is employable. The alpha coefficient was 0.80 at T1 and 0.82 at T2.

Mental well-being was detected using the Italian Mental Health Continuum-Short Form [87] (MHC-SF). The scale consists of 14 items evaluated on a 6-point Likert scale ranging from 1 (never) to 6 (every day) and measures emotional, psychological, and social well-being. An example of an item is "During the past month, how often did you feel happy?". The items were averaged to create a single index. Higher average scores represent higher levels of mental well-being. Cronbach's alpha for the full scale was 0.89 at T1 and 0.92 at T2.

## 3. Data Analysis

Descriptive analyses and Pearson's correlations between variables were computed with SPSS 27 statistical software. The internal consistency of the scales was assessed using Cronbach's alpha coefficient. The hypothesized model (Figure 1) was further tested by performing a path analysis. Through MPLUS 8.7 statistical software, the items were averaged to create a single index. Thus, the predictive value of career ambition (T1), university

reputation (T1), organizational commitment (T1), and technostress (T1) on employability (T2) and well-being (T2) was tested. Furthermore, employability (T2) predictiveness on mental well-being (T2) was assessed. The chi-square ($\chi$2), the Comparative Fit Index (CFI), the Root Mean Square Error of Approximation (RMSEA), and the Standardized Root-Mean-Square Residual (SRMR) were evaluated to determine whether the model had an acceptable fit of the data. A model is usually considered to have an adequate fit when the CFI is at 0.90 or above and the RMSEA or SRMR is at 0.08 or less [88,89]. A good model fit was further indicated by not-significant chi-square ($\chi$2) with values of at least 0.95, and RMSEA and SRMR values lower than 0.06 and 0.08, respectively [90].

*Descriptive Results*

Descriptive statistics and correlations among measures are displayed in Table 1. Career ambition, reputation, and commitment at both T1 and T2 had a significant, positive association with employability and mental well-being measured at both T1 and T2, with r ranging from 0.22 to 0.56, $p < 0.001$. In contrast, technostress at both T1 and T2 was negatively correlated with mental well-being that was assessed at both T1 and T2, with r ranging from $-0.25$, $p < 0.001$, to $-0.12$, $p < 0.05$. Finally, employability had a significant negative correlation with technostress at T1 only: r = $-0.14$, $p < 0.05$.

The hypothesized model was tested and showed a good model fit: $\chi$2 (5) = 5.725, $p = 0.220$, CFI = 0.998, RMSEA = 0.026, CI [0.000, 0.070], SRMR = 0.011. Consistent with our hypotheses, all proposed paths were significant (Figure 2): employability and mental well-being were significantly predicted by the hypothesized variables. More specifically, employability ($R^2 = 0.57$) was significantly predicted by career ambition ($\beta = 0.40$, $p < 0.001$), university reputation ($\beta = 0.22$, $p < 0.001$), university commitment ($\beta = 0.19$, $p < 0.001$), and technostress ($\beta = -0.11$, $p < 0.05$). In addition, career ambition ($\beta = 0.27$, $p < 0.001$), university reputation (0.12, $p < 0.001$), university commitment ($\beta = 0.39$, $p < 0.001$), technostress ($\beta = -0.13$, $p < 0.05$), and employability ($\beta = -0.36$, $p < 0.05$) had a significant impact on mental well-being ($R^2 = 0.29$).

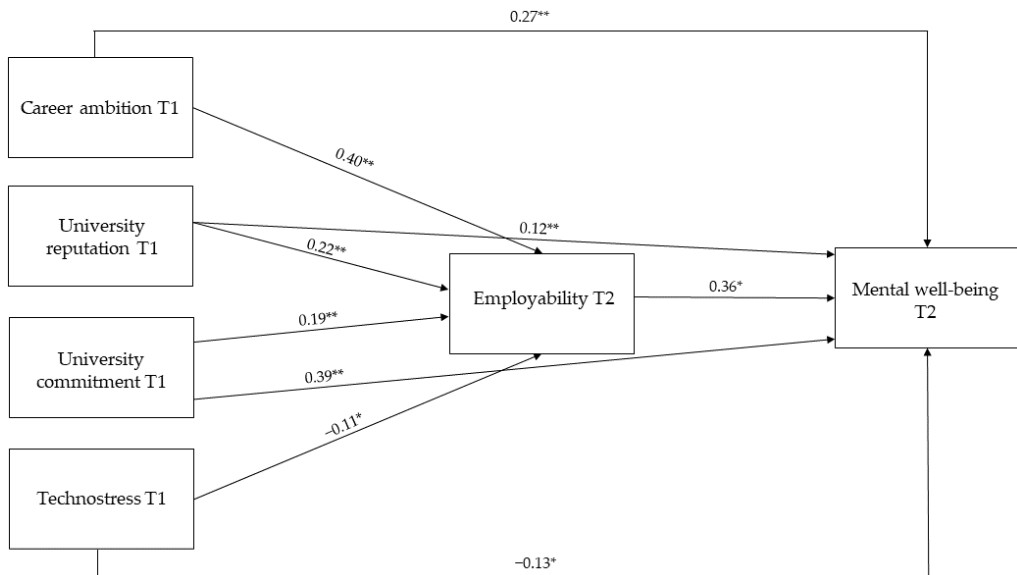

**Figure 2.** Path model with standardized regression coefficients. Note: * $p < 0.05$; ** $p < 0.001$.

**Table 1.** Descriptive statistics and correlations among the psychological variables.

| | M | SD | Ske | Kur | 1 | 2 | 3 | 4 | 5 | 6 | 7 | 8 | 9 | 10 | 11 |
|---|---|---|---|---|---|---|---|---|---|---|---|---|---|---|---|
| 1. Career ambition (T1) | 3.65 | 0.46 | −0.80 | 0.92 | - | | | | | | | | | | |
| 2. University reputation (T1) | 3.59 | 0.81 | −0.45 | 0.05 | 0.23 ** | - | | | | | | | | | |
| 3. University commitment (T1) | 3.25 | 0.56 | −0.55 | 1.44 | 0.33 ** | 0.51 ** | - | | | | | | | | |
| 4. Technostress (T1) | 1.93 | 0.89 | −0.16 | −0.32 | −0.01 | −0.11 | −0.07 | - | | | | | | | |
| 5. Employability (T1) | 3.28 | 0.42 | −0.29 | 2.02 | 0.44 ** | 0.51 ** | 0.50 ** | −0.14 * | - | | | | | | |
| 6. Mental well-being (T1) | 2.40 | 0.82 | −0.12 | −0.17 | 0.33 ** | 0.30 ** | 0.23 ** | −0.25 ** | 0.40 ** | - | | | | | |
| 7. Career ambition (T2) | 3.23 | 0.54 | −0.59 | 0.25 | 0.60 ** | 0.19 ** | 0.15 ** | 0.01 | 0.30 ** | 0.31 ** | - | | | | |
| 8. University reputation (T2) | 3.62 | 0.77 | −0.59 | 0.87 | 0.09 | 0.56 ** | 0.42 ** | −0.09 | 0.37 ** | 0.16 ** | 0.11 * | - | | | |
| 9. University commitment (T2) | 3.23 | 0.59 | −0.37 | 0.78 | 0.21 ** | 0.50 ** | 0.71 ** | −0.08 | 0.40 ** | 0.20 ** | 0.29 ** | 0.50 ** | - | | |
| 10. Technostress (T2) | 2.87 | 0.99 | 0.04 | −0.68 | 0.04 | −0.06 | −0.02 | 0.71 ** | −0.04 | −0.12 * | 0.07 | −0.10 | −0.09 | - | |
| 11. Employability (T2) | 3.28 | 0.42 | −0.18 | 0.89 | 0.36 ** | 0.42 ** | 0.42 ** | −0.11 | 0.71 ** | 0.37 ** | 0.42 ** | 0.52 ** | 0.56 ** | −0.07 | - |
| 12. Mental well-being (T2) | 2.41 | 0.88 | 0.14 | −0.27 | 0.32 ** | 0.27 ** | 0.26 ** | −0.21 ** | 0.30 ** | 0.72 ** | 0.38 ** | 0.22 ** | 0.36 ** | −0.1 6** | 0.46 ** |

Note: * $p < 0.05$; ** $p < 0.001$. M = Mean; SD = Standard Deviation; Ske = Skewness; Kur = Kurtosis.3.2. Path Analysis.

## 4. Discussion

The present study provides an essential contribution to the current literature on psychosocial factors related to students' employability perceptions and mental well-being during the third wave of COVID-19. Italian employment rates for the youngest group of the population, among the lowest in Europe and worsened during the pandemic [1,2]. In line with Event Systems Theory [10], we investigated the factors that shaped how undergraduate students faced changes in universities and the job market due to the health emergency. Thus, we focused on perceived employability as a resource to cope with the uncertainty of the current work environment, especially for young people such as university students. Since university education is a crucial factor in finding a job [7], influencing both employability and mental health, we argued that it could be worthwhile to assess students' perceptions related to their university and career in order to provide useful indications about the design of future interventions aimed at counteracting the decreased employment rates and well-being. Following this reasoning, the study also examined a potential risk factor for employability and well-being (i.e., technostress linked to TEL), which is closely related to the limitation caused by the COVID-19.

As hypothesized, career ambition positively predicted well-being (H1b) and was also a strong predictor of perceived employability (H1a). The strength of such a relationship suggests that students' willingness to achieve their professional goals is a crucial factor in the perception of finding a job in line with their studies. Such a result aligns with previous studies reporting that ambition is an antecedent of proactivity, which, in turn, could predict perceived employability [39]. These findings are of particular relevance since, as far as we know, no study has demonstrated the predictive role of ambition on mental well-being among students during the coronavirus pandemic, suggesting the need to focus on it even more in such a time of crisis. An explanation of this link might be found in Locke and Latham's [39] Goal Setting theory. Locke and Latham's basic assumption is that goals regulate human action, pushing individuals to direct efforts and persistence to accomplish their objectives. As a result, they will be engaged in behaviors that will help them achieve their goals and show greater motivation to participate in such activities. Hence, we could suppose that students with higher scores on ambition have clear career goals, which may be considered an essential factor to realize their career and abilities. Consequently, they might show higher motivation and commitment to study. In turn, this attitude might help them to be more confident in the possibility of finding a job, reducing the anxiety and uncertainty related to the labor market at the time of Coronavirus [29]. The predictive role of employability in mental well-being is consistent with this explanation (H5). Students who are higher in employability experienced more increased mental well-being, supporting our thesis that employability, even more so in the pandemic, has been a crucial resource in facing stressors such as the rapid and unexpected changes characterizing work and academic education. Lo Presti et al. [91] found that employability activities mediated the association between career competencies and subjective career success. This study does not represent the exact process that we propose, but our results and the empirical evidence from it open a possibility for future research to consider. Perceptions of employability could be considered an important mediator between career dimensions and mental and occupational well-being of future workers. This is another important reflection for stakeholders and for rethinking a sustainable working world.

In the pandemic outbreak, the academic education has been provided in remote learning mode. Consistently with literature [70,81], technostress related to TEL was found to be a risk factor for perceived employability (H4a) and mental well-being (H4b). It must be noted that most participants are digital natives [92] because they were born in a world connected to the internet, and ICTs are part of their daily routines [93]. Therefore, on average, they have more digital skills than individuals born in the years preceding the new millennium. Still, despite this, they experienced a negative impact of technostress on their well-being and career. These data support the literature that has underlined that students attending online courses experience higher technostress than those enrolling in

face-to-face mode [66]. As posited by the P–E fit theory [73], stress is the result of an unbalanced relationship between an individual and the environment. Thus, the balancing between resources and demands, both of universities and students, seems to have not been adequate to cope effectively with the remote learning challenge. This mismatch may have led our participants to believe they could not carry out development activities, such as completing their university studies, perceived as fundamental to obtaining a job. As a result, they perceived decreased employability and mental well-being. Apart from the above-mentioned factors, the great importance placed by students on their learning contexts to realize their abilities and aspirations was further revealed by the confirmation of H2 and H3. The findings showed that perceiving one's university as a well-reputed organization increases both perceived employability (H3a) and mental well-being (H3b). These relationships are consistent with a large amount of research [41–43] reporting how high-ranked universities attract students with better skills and are more likely to offer employability skills and job-related opportunities. Thus, it is plausible to suppose that students may have been perceived to be more employable (H2a) and experienced higher mental well-being (H2b) because they had recognized the importance of these activities in achieving career goals and, more in general, for their lives. These findings support, also among Italian students in the Coronavirus era, the conclusion of Pitan and Muller's [22] study in a prepandemic time and South African academic context, namely that it is both students' personal capital (such as skills acquired through learning) and their external conditions (such as the reputation of the university) that shape their perception of being employable.

Regarding H3, employability and well-being were significantly and positively predicted by university commitment. This result is consistent with previous studies [29,52] highlighting the positive individual and organizational outcomes linked to employees' identification and emotional attachment to their organization. The feeling of being part of the organization satisfies socio-emotional needs such as affiliation, esteem, and emotional support [55]. Therefore, we could imagine that students' commitment to their university could have acted as a buffer against COVID-19's unfavorable outcomes on perceived employability and mental well-being. Moreover, it must be noted that, even though students' bond to their university was potentially at risk as a result of TEL adoption, the participants reported above-average scores (Table 1) on the commitment to their university. Such data might be interpreted as the students' possibility to have access to valued resources to face the demands of the pandemic challenges thanks to the strong bond with their universities.

*Limitations and Future Research Design*

The current study had several limitations that should be taken into account. First of all, only self-report measures were employed. Then, we could not rule out the impact of shared method bias, which may have exacerbated the observed associations between the variables. Furthermore, the Ambition Scale yielded a low internal consistency at T1 (0.62). However, given that all other scales had acceptable alphas, the value was not considered as threatening to the validity of the study results [94,95]. A further weakness is the lack of representativeness of the sample. Due to the adoption of convenience sampling, we cannot claim that the results are generalizable to the Italian population. Thirdly, the research did not consider other outcomes that could have been affected by the analyzed factors, such as academic performance. Finally, we cannot rule out that the relationships investigated in the present study are somehow bidirectional (e.g., high levels of perceived employability may affect high levels of university) or that there are other possible antecedents of employability and well-being that we have not included in the model. Thus, an interesting avenue for further studies entails the examination of the potential predictiveness role of individuals' academic efficacy in developing perceived employability, as investigated in prepandemic studies [96].

## 5. Conclusions

Despite the limitations highlighted above, this study provided valuable new insights on which factors might be involved in undergraduates' perceptions about their employability and well-being, filling a gap in the literature. At the time of writing, no research has examined the predictiveness of ambition, university reputation, and university commitment in university students' perceived employability and well-being in Italy. Moreover, to our knowledge, no study has ever investigated technostress related to remote learning as a predictor of employability among students in both pandemic and prepandemic times. Our findings suggest conceiving employability as pivotal in individuals' well-being during COVID-19. Additionally, the identification of employability antecedents highlighted that students who are willing to commit to achieving their professional goals were more efficient in adapting to the uncertainty of the labor market, experiencing confidence in the possibility of finding a job in line with their aspiration even during the health emergency. This belief is related to a sort of perceived control over one's career, which may play a protective role against adverse outcomes caused by the uncertainty of the labor market, especially in the pandemic and for young people. Thus, it would be reasonable to presume that, as a result of such perceived control, students have been able to cope better with changes imposed by the coronavirus both on the job market and in the educational context (i.e., the adoption of TEL).

Furthermore, the extent to which students' rating of their university and the strength of the bond with their university are crucial in triggering such an adaptation process supports the thesis [22] that perceived employability is jointly determined by both individual and external characteristics. In particular, university reputation during the crisis could have represented a stabilizing element in an unstable employment environment for young people, fostering, in such a way, the mentioned perception of control.

In addition to the above evidence, the results underline the need to pay serious attention to a potential stressor related to the forced transition to face-to-face to TEL in the health emergency: technostress. The latter seems to have shaped unfavorable outcomes of students' beliefs on their professional careers as well as on their well-being. Hence, it is desirable that institutions and academic organizations seriously need to conduct more work in this area to mitigate the sharply increasing unemployment due to the COVID-19. In order to achieve such an aim, our study sheds light on the need for tailored academic interventions that could be implemented to target, either separately or jointly, ambition, university commitment, reputation, and technostress. These interventions may focus on the following two fronts.

On the one hand, a greater effort to increase universities reputation is essential. In this regard, as argued by Ma [97], investing in visibility can improve organizational reputation. This goal, in turn, can be pursued by adopting effective external communication, for example, through the most used communication media by young people: social networks [82]. In this way, universities may attract students and graduates both at a national level and an international one and, in a sort of chain reaction, increase their reputation.

On the other hand, in a complementary way, political authorities and universities should invest in improving employability skills provided to students, in line with the literature [98] that recommends the need to reproject universities' curricula and introduce new methodologies to improve students' sustainable employability. This might translate into not only better job performances and job opportunities but also more efficient coping with job-related and educational-related stressors (e.g., technostress associated with TEL).

**Author Contributions:** Conceptualization, G.S., L.M. and V.C.; Formal analysis, G.S., L.M. and V.C. Investigation, G.S., L.M. and V.C.; writing—original draft preparation, G.S., L.M. and V.C.; writing—review, G.S., L.M. and V.C. All authors have read and agreed to the published version of the manuscript.

**Funding:** This research received no external funding.

**Institutional Review Board Statement:** The study protocol was conducted in accordance with APA and University Federico II ethical standards. In accordance with the provisions of Italian law, since there was no treatment of persons, no authorization was required from the ethics committee, but it was only necessary to follow the rules proposed by it (see link at: http://www.comitatoeticofedericoii.it, accessed on 15 September 2019). The study conformed to the ethical principles of the 1995 Helsinki Declaration.

**Informed Consent Statement:** Informed consent was obtained from all individual participants included in the study.

**Data Availability Statement:** The data presented in this study are available on request from the corresponding author.

**Conflicts of Interest:** The authors declare no conflict of interest.

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
