# Peer review of "The Impact of University-Related Variables on Students’ Perceived Employability and Mental Well-Being: An Italian Longitudinal Study"

_sustainability, doi:10.3390/su14052671_

Round 1

Reviewer 1 Report

  1. This is a good study.
  2.  Indicate the response rate in method chapter. 
  3. Add in brief description on scoring method for each instrument used for the study. 

Author Response

Dear Reviewer,

Thanks for the feedback on our manuscript. The latter has been revised according to the suggestions. In the attached file, there are the responses to the comments.

Thanks again

Reviewer 2 Report

Dear authors,
first of all, thank you for the opportunity to review your article, which I found very interesting. The study refers to the impact that the pandemic had, in terms of personal and organisational antecedents, on students' employability and mental health. In addition, your study is longitudinal in nature, thus overcoming some of the limitations typical of cross-sectional surveys. I carefully read your work, and I have a few minor revisions to improve its thoroughness. Here are my suggestions:

1) Could the measurement time reference also be included in Figure 1? 
2) Line 210: Is there a mistake? First you say that 301 students attended the course and that you asked them to participate in the survey, but then in T1 you talk about 550 individuals. 
3) Could you insert the drop-out rate between T1 and T2?
4) Line 258: "Data analysis" I think it is a subsection (in bold?).
5) Table 1: could you also include asymmetry and kurtosis indices? Also in my opinion it is not necessary to put the 1's diagonally on the correlation matrix, a simple dash - would suffice.
6) You have considered Cronbach's alpha as a measure of reliability. You should enter the cutoff that you deemed as valid, since an index is below 0.70. You have rightly reported it within the limits, however. I would point out that there are studies, nevertheless, that indicate that in certain circumstances even the .60 cutoff can be accepted (Chretien, J. L., Nimon, K., Reio Jr, T. G., & Lewis, J. (2020). Responding to Low Coefficient Alpha: Potential Alternatives to the File Drawer. Human Resource Development Review, 1534484320924151).
7) I also recommend evaluating the convergent (Mean Variance Extracted) and discriminant (Maximum Shared Variance) validity of latent constructs, to make the measurements used more robust.
8) Although it is not explained in the paper, considering the paragraph "Path analysis" I imagine that you have collapsed the different items into a single measure for the computation of the model. Did you use the average or the sum of the scores? This aspect should be clarified. 
9) Lines 275-280: Am I wrong or is the font smaller than the rest of the text?
10) Line 295: . * p < 0.005, I think it is < 0.05
11) Another interesting aspect of your work is that indeed employability mediates the relationships you hypothesized. In my opinion you should mention and give importance to this finding.

Author Response

Dear Reviewer,

Thanks for your valuable contribution. Your feedback has been used to revise the manuscript. The responses to the comments are listed in the attached file.

Best regards

Reviewer 3 Report

Dear Authors,

Thank you for your valuable contribution to up to date and important topic – perceived employability and mental well-being.

Although the subject literature addressing employability and mental well-being is broad, there is a need for further scientific exploration.

The paper is well written. From the formal side, the article does not raise any major reservations.

Please consider the inclusion in the article title “perceived employability” and “mental well-being”.

According to the literature review I recommend to discuss other definitions of perceived employability. This article below are focused on perceived employability and organizational learning. Be so kind and consider:

WiÅ›niewska, S.; WiÅ›niewski, K.; SzydÅ‚o, R. The Relationship between Organizational Learning at the Individual Level and Perceived Employability: A Model-Based Approach. Sustainability 2021, 13, 7561. https://doi.org/10.3390/su13147561 

It is necessary to refer to sustainability, which should be discussed in the article.

In the bibliographic reference lists the name of the journal must be written with its abbreviation, for example, in 27 should be “Electron. J. Appl. Stat. Anal.”, in 45 “Hum. Resour. Manage.”, in 50 should be “Bangladesh J. Multidiscip Sci. Res.”, in 76 should be “Int. J. Environ. Res. Public.”.

I hope that my suggestions and comments will be helpful.

Author Response

Dear Reviewer,

Thanks for the helpful suggestions. We have revised the manuscript based on the suggestions. In the attached file, there are our responses to the comments.

Best regards
